# Allogenic Synovia-Derived Mesenchymal Stem Cells for Treatment of Equine Tendinopathies and Desmopathies—Proof of Concept

**DOI:** 10.3390/ani13081312

**Published:** 2023-04-11

**Authors:** Inês Leal Reis, Bruna Lopes, Patrícia Sousa, Ana Catarina Sousa, Mariana Branquinho, Ana Rita Caseiro, Sílvia Santos Pedrosa, Alexandra Rêma, Cláudia Oliveira, Beatriz Porto, Luís Atayde, Irina Amorim, Rui Alvites, Jorge Miguel Santos, Ana Colette Maurício

**Affiliations:** 1Departamento de Clínicas Veterinárias, Instituto de Ciências Biomédicas de Abel Salazar (ICBAS), Universidade do Porto (UP), Rua de Jorge Viterbo Ferreira, No. 228, 4050-313 Porto, Portugal; lealreines@gmail.com (I.L.R.); brunisabel95@gmail.com (B.L.); pfrfs_10@hotmail.com (P.S.); anacatarinasoaressousa@hotmail.com (A.C.S.); m.esteves.vieira@gmail.com (M.B.); alexandra.rema@gmail.com (A.R.); ataydelm@gmail.com (L.A.); ruialvites@hotmail.com (R.A.); jmposs1970@gmail.com (J.M.S.); 2Centro de Estudos de Ciência Animal (CECA), Instituto de Ciências, Tecnologias e Agroambiente da Universidade do Porto (ICETA), Rua D. Manuel II, Apartado 55142, 4051-401 Porto, Portugal; rita.caseiro.santos@gmail.com (A.R.C.); s.santospedrosa@gmail.com (S.S.P.); 3Associate Laboratory for Animal and Veterinary Science (AL4AnimalS), 1300-477 Lisboa, Portugal; 4Cooperativa de Ensino Superior Politécnico e Universitário (CESPU), Avenida Central de Gandra 1317, 4585-116 Gandra, Portugal; 5University School Vasco da Gama (EUVG), Avenida José R. Sousa Fernandes, 3020-210 Coimbra, Portugal; 6Vasco da Gama Research Center (CIVG), University School Vasco da Gama (EUVG), Avenida José R. Sousa Fernandes, 3020-210 Coimbra, Portugal; 7Centro de Biotecnologia e Química Fina (CBQF), Laboratório Associado, Escola Superior de Biotecnologia, Universidade Católica Portuguesa, Rua de Diogo Botelho 1327, 4169-005 Porto, Portugal; 8Laboratório de Citogenética, Instituto de Ciências Biomédicas de Abel Salazar (ICBAS), Universidade do Porto (UP), Rua de Jorge Viterbo Ferreira, No. 228, 4050-313 Porto, Portugal; csoliveira@icbas.up.pt (C.O.); bporto@icbas.up.pt (B.P.); 9Departamento de Patologia e Imunologia Molecular, Instituto de Ciências Biomédicas de Abel Salazar (ICBAS), Universidade do Porto (UP), Rua de Jorge Viterbo Ferreira, No. 228, 4050-313 Porto, Portugal; ifamorim@icbas.up.pt; 10Instituto de Investigação e Inovação em Saúde (i3S), Universidade do Porto (UP), Rua Alfredo Allen, 4200-135 Porto, Portugal

**Keywords:** allogenic, cell-based therapies, clinical trials, ligament, mesenchymal stem cells, sport horses, synovial mesenchymal stem cell, tendon

## Abstract

**Simple Summary:**

Horses are high-level athletic athletes prone to musculoskeletal injuries. Tendon/ligament injuries are the most frequent types of injuries which that are very difficult to treat. Instead of tissue regeneration, usually, fibrous scar tissue develops which leads to decreased functionality of the injured area and threatens the participation of sport horses. The aim of regenerative medicine is to find a treatment that promotes tissue regeneration and that allows the equine patient to return to the same level of athletic performance in the shortest time period possible. In this study, we developed a solution of equine synovial membrane stem cells and autologous serum, to be injected at the lesion site to promote tissue regeneration. We describe the processes of tissue collection, preparation, isolation of synovial stem cells, expansion, culture, cryopreservation, and posterior preparation with autologous serum. The solution was tested in 16 tendons and ligaments of equines. After treatment, all equine patients underwent a physical rehabilitation program and were monitored with physical and ultrasonographic exams. The results were very promising, and thus, support the use of equine synovial stem cells and autologous serum in the treatment of tendonitis and desmitis.

**Abstract:**

Tendon and ligament injuries are frequent in sport horses and humans, and such injuries represent a significant therapeutic challenge. Tissue regeneration and function recovery are the paramount goals of tendon and ligament lesion management. Nowadays, several regenerative treatments are being developed, based on the use of stem cell and stem cell-based therapies. In the present study, the preparation of equine synovial membrane mesenchymal stem cells (eSM-MSCs) is described for clinical use, collection, transport, isolation, differentiation, characterization, and application. These cells are fibroblast-like and grow in clusters. They retain osteogenic, chondrogenic, and adipogenic differentiation potential. We present 16 clinical cases of tendonitis and desmitis, treated with allogenic eSM-MSCs and autologous serum, and we also include their evaluation, treatment, and follow-up. The concerns associated with the use of autologous serum as a vehicle are related to a reduced immunogenic response after the administration of this therapeutic combination, as well as the pro-regenerative effects from the growth factors and immunoglobulins that are part of its constitution. Most of the cases (14/16) healed in 30 days and presented good outcomes. Treatment of tendon and ligament lesions with a mixture of eSM-MSCs and autologous serum appears to be a promising clinical option for this category of lesions in equine patients.

## 1. Introduction

Tendonitis and desmitis are defying clinical challenges in equine patients that require long recovery periods, and ineffective tendon repair can affect their sport careers. Tendons operate near their functional limits during maximal exercise, and their ability to adapt to stress and self-repair is limited. A controlled exercise program alone or in combination with a variety of conservative treatments, such as corrective shoeing and nonsteroidal anti-inflammatory drugs (NSAIDs), is still the gold standard therapy for equine tendon disease [1]. Current treatments often do not fully repair or regenerate the injured or affected tendon nor lead to its total functional recovery [1,2].

The aim of tendinopathy treatment is to achieve tissue regeneration and return to complete organ function and performance. Recently, tissue engineering approaches have attracted attention for tissue repair. Among the approaches, the use of mesenchymal stem cell-based therapy has increased, since it is a promising approach for tissue repair and regeneration including tendinopathy and desmitis [1,3,4,5,6].

Mesenchymal stem cells (MSCs) can be isolated from several tissue sources such as bone marrow, peripheral blood, dental pulp, umbilical cord, and amniotic fluid [7]. MSC characteristics have been defined by the Mesenchymal and Tissue Stem Cell Committee of the International Society for Cellular Therapy (ISCT), and include plastic adherence when maintained in standard culture conditions; expressing clusters of differentiation (CDs), such as CD44, CD90, and CD105; and no expression of major histocompatibility complex (MHC)-class II markers and of hematopoietic-related markers (CD45 and CD34) [8]. Finally, MSCs must be able to differentiate in vitro into, at least, osteoblasts, adipocytes, and chondroblasts, in the presence of adequate differentiation culture media [8].

Synovial membrane mesenchymal stem cells (SM-MSCs) were initially isolated, in 2001, by De Bari et al. [9], from human knee joints and showed significant proliferative ability in culture, even after Passage 10 (P10), and multilineage differentiation potential in vitro [9]. These cells represent a good source of MSCs and a promising therapeutic tool mostly for musculoskeletal pathologies [10]. Sakagushi et al. compared the properties of different sources of human stem cells, i.e., bone marrow, synovium, periosteum, skeletal muscle, and adipose tissue, and observed the superiority of synovium as a source for MSCs for treatment of musculoskeletal pathologies as they had more ability to chondrogenesis. Pellets of synovium-derived stem cells were larger and expressed more intense staining for chondrogenic differentiation [11].

SM-MSCs have higher chondrogenic capacity than other studied sources of MSCs, such as bone marrow (BM-MSCs) [12,13]. Cartilage pellets from SM-MSCs have been reported to be significantly larger than those from BM-MSCs [12]. SM-MSCs have a higher production of uridine diphosphate glucose dehydrogenase (UDPGD) [13], an enzyme that converts UDP-glucose into UDP-glucuronate, one of the two substrates required by hyaluronan synthase for hyaluronan polymer assembly. In addition, Sox-9, collagen type II (Col-II), and aggrecan, specific markers for chondrogenesis, as well as cartilage-specific molecules such as cartilage oligomeric matrix protein (COMP) have also been found in high amounts in equine synovial fluid-derived MSCs and the extracellular matrix, respectively by reverse transcription polymerase chain reaction (RT-PCR) [13].

In a recent study, using a rabbit model, Bami et al. highlighted the superiority of SM-MSCs in terms of chondrogenesis, osteogenesis, myogenesis, and tenogenesis [14]. A study of xenogenic implantation of SM-MSCs in equine articular defects also confirmed better healing of the cartilage of affected knees as well as a higher expression of collagen type II, indicating the presence of hyaline cartilage in the healed defect [15].

SM-MSCs have been defined as MSCs due to their phenotypic profile and differentiation potential. Even though there are no specific antibody markers to identify these MSCs, there is general agreement that MSCs should be negative to hematopoietic markers CD34 and CD45 and positive to CD44, CD73, CD90, and CD105 [16]. Mochizuki et al. found that SM-MSCs maintained their proliferative ability, regardless of which region they were collected from in the synovium [17].

In 2003, Fickert et al. reported that the markers CD9, CD44, CD54, CD90, and CD166 could be used to identify MSCs isolated from the synovium of human patients with osteoarthritis (OA), and they also confirmed that CD9/CD90/CD166 triple-positive cell subgroups had obvious chondrogenic and osteogenic differentiation abilities [18].

Prado et al. confirmed the mesenchymal nature of equine synovial membrane and fluid-derived stem cells through the expression of significant hematopoietic (CD45, CD34, CD117, and CD133), mesenchymal (CD105, CD90), pluripotency (OCT3/4 and NANOG), embryonic (Tra-1-81), inflammatory, and angiogenesis (vascular endothelial growth factor (VEGF-R1) and LY6a) markers [19]. Although the presence of hematopoietic and inflammatory markers was not expected, variations may occur and must be considered to influence acute or chronic stages of osteochondrosis expression and/or inflammatory events [19,20].

Nevertheless, the immunophenotype characterization of equine MSCs (eMSCs), as well as in other veterinary species, has not yet been completely established [19]. This is a major challenge, since the expression of certain adult stem cell markers may differ between species. For that reason, it is a need to define a set of CD markers which can be uniformly applied for the identification of eMSCs [8,20].

Horses are high performance athletes prone to musculoskeletal diseases, i.e., osteoarticular, as well as tendon/ligament lesions and fractures of various degrees due to sport- and age-related injuries. These pathologies resemble human musculoskeletal conditions, and therefore, horses are a valuable animal model for assessing stem cell and cell-based therapies prior to the translation of results into humans [21]. The use of a therapy that can regenerate these structures and restore their complete functionality instead of ordinary healing is the aim of our study and of equine practitioners throughout the world. 

Recent studies have suggested that MSCs can self-renew, migrate to injury sites (homing), perform multilineage differentiation, and secrete bioactive factors, thus, increasing proliferation and migration of tendon stem/progenitor cells via paracrine signaling and increasing the regeneration ability of tissues with poor aptitude [1,3,4,5,22,23].

In fact, the knowledge of the importance of this paracrine action has opened doors to cell-free therapeutic strategies in regenerative medicine. The soluble factors (cytokines, chemokines, and growth factors) and nonsoluble factors (extracellular vesicles and exosomes) released in the extracellular space by MSCs, commonly known as secretome, have become the focus of novel therapeutic approaches due to their key role in cell-to-cell communication, their active influence on immune modulation, and their pro-regenerative capacity both in vitro and in vivo [23]. Therefore, in this study, secretome was also analyzed with the prospect of being used therapeutically, in the future, in similar clinical cases.

In the present study, equines used as show jumping and dressage athletes as well as leisure horses with acute and chronic lesions were treated with intralesional administration of the considered combination, i.e., autologous serum and eSM-MSCs. The treatment consisted of two injections, 15 days apart. Pre- and post-treatment evaluations consisted of clinical, orthopedic, and tendon/ligament ultrasound exams. None of the selected equine patients had previously received any other regenerative treatment.

## 2. Materials and Methods

### 2.1. Study Design and Horse Selection

This prospective longitudinal study was performed in Portugal between February 2016 and January 2019. Sixteen horses, from 5 to 22 years old with acute and chronic signs of lameness were enrolled in this study (11 males and 5 mares), whose sport activities were distributed over show jumping (14), dressage (1), and leisure (1). The horses were all outpatients from an equine ambulatory clinic. This study included the treatment of 16 tendons, i.e., 14 superficial digital flexor tendons and 2 deep digital flexor tendons, and 4 suspensory ligaments. 

Lameness was scored based on the American Association of Equine Practitioners (AAEP) scale (Table 1) and confirmed by using a positive regional nerve block. Flexion and pain to pressure tests were also evaluated [24]. 

### 2.2. Inclusion and Exclusion Criteria

In this study, horses with acute or chronic lameness (Table 2), with diagnosed tendonitis and/or desmitis and with no signs of systemic disease were accepted in the inclusion criteria. Injured horses were treated in acute stages of disease, except for two equine patients (Patients 3 and 6). Patient 3 had an injury the year before this treatment and laser therapy had been performed, without a complete recovery. After that, he had a re-injury and, at this time, this treatment was suggested. Patient 6 was referred by another clinician who tried, unsuccessfully, to treat this patient. Patient 6 was sent to the field for one year and then re-evaluated. At this time, and as its trainer wanted to improve its quality of life, this treatment was proposed by its clinician. The lameness grade of each equine patient is specified in Table 2. Considering the established exclusion criteria, the selected equine patients should not have been under any other medical treatment (including nonsteroidal anti-inflammatory drugs, intra-articular corticosteroids, hyaluronan, glycosaminoglycans, platelet-rich plasma (PRP), and other MSC preparations) for at least 2 months before the allogenic eSM-MSC treatment and did not receive any additional medical treatment (except for that described in the treatment plan) for at least 2 months post the cell-based treatment.

### 2.3. Ethics and Regulation

This study was carried out in accordance with the Organismo Responsável pelo Bem Estar Animal (ORBEA) from ICBAS-UP, project number P289/ORBEA/2018 recommendations and authorization. Treatments were performed with permission and signature of an informed consent from the equine patient’s legal trainer, following a thorough explanation on the procedure itself and possible risks and associated effects, in accordance with national regulations and project approval from the competent authorities. In addition, no animals were euthanized for this study.

### 2.4. Donor Selection and SM Collection

The eSM-MSC donor was a young and healthy foal, 7 months old, who died accidentally when running in the arena. The trainer authorized synovial membrane collection from the hocks, knees, and fetlocks. The synovial membrane was evaluated and its appearance was transparent, bright, and smooth; in addition, the presence of viscous and transparent synovial fluid confirmed its soundness. The skin covering the incisional field was surgically cleaned with chlorohexidine and alcohol. The skin and subcutaneous tissue were incised, debrided, the articular capsule was opened, and the synovial membrane was isolated and extracted into a Dulbecco′s phosphate buffered saline (DPBS) container. The samples were transported to the laboratory with ice packs for refrigerated temperatures. Figure 1a presents the fresh tissue arrival and Figure 1b shows the preparation at the laboratory. Figure 2 shows a schematic representation of the process from eSM-MSC collection to the administration of the combination, i.e., eSM-MSCs and autologous serum (1 × 10^6^ cells/mL and 1 mL of autologous serum in a total volume of 2 mL).

### 2.5. eSM-MSC Isolation

After collection, the equine synovial membrane was prepared at the Laboratory of Veterinary Cell-based Therapies from ICBAS-UP. The isolation protocol for eSM-MSCs was developed by patented proprietary technology Regenera^®^ (PCT/IB2019/052006, WO2019175773, Compositions in use for the treatment of musculoskeletal conditions and methods for producing the same leveraging of the synergistic activity of two different types of mesenchymal stromal/stem cells, Regenera^®^). Fresh tissue was transported to the laboratory facilities in a hermetically sealed sterile container in transport media (supplemented with 3% (*v*/*v*) penicillin-streptomycin (Gibco^®^, Waltham, MA, USA) and 3% amphotericin B (Gibco^®^) and processed within a period of up to 48 h. The synovial tissue was digested using collagenase and the isolated cells were incubated in a static monolayer culture using standard MSC basal medium supplemented with 10% fetal bovine serum (FBS) and maintained in standard culture conditions (37 °C, 5% CO_2_, and humidified atmosphere) until they reached confluence. Cells from confluent cultures were cryopreserved in 10% dimethylsulphoxide (DMSO) and FBS, at a concentration of 3 × 10⁶ cells/mL, using a control rate temperature freezer (Sy-Lab Cryobiology, SY-LAB Geräte GmbH, Purkersdorf, Austria). For expansion optimization, cells were cryopreserved in passages (P) between P2 and P3 to generate suitable master cell banks (MCBs). Expansion, thereafter, was analyzed during a maximum of 20 cumulative population doublings (cCPDs). The range of cCPDs chosen allowed for enough expansion to maximize the number of cells in the working cell banks (WCB) but kept the cCPDs within the genomic stability range.

### 2.6. SM-MSC Characterization

#### 2.6.1. Tri-Lineage Differentiation Protocols

For all the differentiation protocols, cells in P4 were used after thawing.

##### Adipogenic Differentiation and Oil Red O Staining

For the adipogenic differentiation protocol, 1 × 10^4^ cells/cm^2^ were seeded in the wells of a 12-well plate (cell culture plates, 12-well, VWR^®^, Suwanee, Atlanta, GA, USA), with the addition of the standard culture medium. The plate was incubated under standard conditions for 4 days. After this period, the culture medium of 10 wells was replaced by complete adipogenesis differentiation medium (StemPro^®^ Adipogenesis Differentiation Kit, Gibco^®^, Waltham, MA, USA), 2 wells were used as controls and maintained with the standard culture medium. Following the manufacturer’s instructions, the media were replaced every 3–4 days and the cells maintained in differentiation for 14 days. At the end of this period, the oil red O staining protocol was performed using a handmade solution. The culture differentiation medium was removed, and the wells were gently washed with PBS. Cells were fixed with 4% formaldehyde (3.7–4% buffered to pH 7, reference# 252931.1315, Panreac AppliChem^®^, Darmstadt, Germany) for 10 min at room temperature, and the wells were washed 3 additional times with phosphate-buffered saline (PBS). Oil red O solution was added to each well and the plate incubated for 10–20 min at room temperature. Oil red O was discarded, and any excess dye was removed by several washes with PBS. PBS was added to each well for visualization. The aim of this assay was the identification of rounded cells with intracytoplasmic lipid vacuoles and their red coloration due to the exposure to the oil red O solution.

##### Chondrogenic Differentiation and Alcian Blue Staining

Thawed eSM-MSCs were automatically counted, and cell viability determined (%). Then, the cells were centrifuged, supernatant removed, and the pellet resuspended in culture medium to generate a cell suspension with 1.6 × 10^7^ viable cells/mL. To generate micro-mass cultures, 5 μL droplets of the cell suspension were placed in the center of 10 wells of a 96-well plate (cell culture plates, 96-well, VWR^®^, Suwanee, Atlanta, GA, USA) to induce chondrogenic differentiation. The plate was maintained under standard conditions for 2 h. After this time, chondrogenic differentiation medium (StemPro^®^ Chondrogenesis Differentiation Kit, Gibco^®^, Waltham, MA, USA) was added to 8 wells; the other 2 wells were considered to be controls and to these, the standard culture medium was added. Following the manufacturer’s instructions, the media were replaced every 3–4 days and cells maintained in differentiation for 14 days. At the end of this period, the Alcian blue staining, pH 2.5, protocol was performed (Alcian Blue 8GX, Sigma-Aldrich^®^, St. Louis, MO, USA). The culture differentiation medium was removed, and the wells were gently washed with PBS. The cells were fixed with 4% formaldehyde for 20 min at room temperature, and the wells were washed 3 additional times with PBS. Alcian blue solution was added to each well and the plate incubated for 30 min at room temperature. Then, the Alcian blue was discarded, and the wells were rinsed 3 times with 3% acetic acid (*v*/*v*). For neutralization of acidity and for visualization by inverted phase contrast microscopy, distilled water was added to all wells. The aim of this assay was the identification of chondrogenic aggregates and their coloration in blue due to the exposure to Alcian blue solution.

##### Osteogenic Differentiation and Alizarin Red Staining

For osteogenic differentiation, 8 × 10^3^ cells/cm^2^ were seeded into the wells of a 12-well plate. The plate was maintained under standard conditions for 4 days. After this period, the culture medium of 10 wells was replaced by complete osteogenic differentiation medium (StemPro^®^ Osteogenic Differentiation Kit, Gibco^®^, Waltham, MA, USA), and 2 wells were used as controls and maintained with the standard culture medium. Following the manufacturer’s instructions, the media were replaced every 3–4 days and the cells maintained in differentiation for 21 days. At the end of this period, the alizarin red s staining protocol was performed using a commercial solution (alizarin red staining solution, Milllipore^®^, Burlington, MA, USA). The culture differentiation medium was removed, and the wells were gently washed with PBS. The cells were fixed with 4% formaldehyde for 30 min at room temperature, and the wells were washed twice with distilled water. One ml of 40 mM of alizarin red solution was added to each well and the plate incubated for 30 min. Then, the alizarin red solution was discarded, and the wells were rinsed 3 times with distilled water until the supernatant became clear. For visualization by inverted phase contrast microscopy, PBS was added to all the wells. The aim of this assay was to identify calcium-containing osteocytes stained in red after exposure to alizarin red solution. 

#### 2.6.2. Karyotype Analysis

The eSM-MSCs in two different passages (P4 and P7) were submitted to cytogenetic analysis to determine the genetic stability in terms of chromosome number and occurrence of neoplastic changes. For both passages, 70–80% confluence was reached. Then, the culture medium was changed and supplemented with 10 μg/mL colcemid solution (KaryoMAX^®^ Colcemid™ Solution, Gibco^®^, Waltham, MA, USA). After 4 h, the eSM-MSCs were collected and resuspended in 8 mL of 0.075 M KCl solution, followed by incubation under standard conditions for 15 min. After centrifugation (1700 rpm), 8 mL of ice-cold fixative comprising methanol and glacial acetic at a proportion of 3:1 was added and mixed. Afterwards, the cells were centrifuged again. Three fixation rounds were carried out. After the last centrifugation, the suspension of eSM-MSCs was spread over glass slides. A karyotype analysis was performed by one scorer on Giemsa-stained cells. For the different passages, a specific number of cells in metaphase were evaluated depending on the number of cells with a normal karyotype identified, guaranteeing a better representation of the population under study.

#### 2.6.3. Secretome Cell Conditioned Medium (CM) Analysis

The eSM-MSCs were harvested from equine synovial membrane and maintained in culture, as previously described. The cells in P4 were subjected to an analysis of their conditioned medium (CM) to identify cytokines and chemokines secreted after conditioning. When in culture, after reaching a confluence of around 70–80%, the culture medium was removed, and the culture flasks were gently washed with DPBS two to three times. Then, the culture flasks were further washed two to three times with the basal culture medium of each cell type, without any supplementation. To begin the conditioning, non-supplemented DMEM/F12 GlutaMAX™ (10565018, Gibco^®^, Thermo Fisher Scientific^®^, Waltham, MA, USA) culture medium was added to the culture flasks, which were then incubated under standard conditions. The culture medium rich in factors secreted by the cells (CM) was collected after 48 h. The collected CM was then concentrated five times. After collection, it was centrifuged for 10 min at 1600 rpm, and its supernatant collected and filtered with a 0.2 μm syringe filter (Filtropur S^®^, PES, Sarstedt, Nümbrecht, Germany). For the concentration procedure, Pierce™ Protein Concentrator, 3k MWCO, 5–20 mL tubes (88525, Thermo Scientific^®^, Waltham, MA, USA) were used. Initially, the concentrators were sterilized following the manufacturer’s instructions. Briefly, the upper compartment of each concentrator tube was filled with 70% ethanol (*v*/*v*) and centrifuged at 300× *g* for 10 min. At the end of the centrifugation, the ethanol was discarded, and the same procedure was carried out with DPBS. Each concentrator tube was subjected to two such centrifugation cycles, followed by a 10 min period in the laminar flow hood to complete drying. Finally, the upper compartment of the concentrator tubes was filled with plain CM (1 × concentration) and subjected to new centrifugation cycles, under the conditions described above, for the number of cycles necessary to obtain the desired CM concentration (5×). The concentrated CM was stored at −20 °C and subsequently subjected to a Multiplexing LASER Bead analysis (Eve Technologies, Calgary, AB, Canada) to identify a set of biomarkers present in the Equine Cytokine 8-Plex Assay (EQCYT-08-501). The list of searched biomarkers includes basic fibroblast growth factor (FGF-2), granulocyte colony-stimulating factor (G-CSF), granulocyte macrophage colony-stimulating factor (GM-CSF), monocyte chemoattractant protein-1 (MCP-1), interleukins (IL) IL-6, IL-8, IL-17A, and human growth-regulated oncogene/keratinocyte chemoattractant (GRO/KC). All samples were analyzed in duplicate.

#### 2.6.4. Immunohistochemistry

Early passages of eSM-MSCs-P0 and -P3 were maintained in culture until a confluence of 70–80% was reached, and then enzymatic detachment was performed with 0.25% trypsin-EDTA solution. A cytoblock was performed fixing the cells with Sure Thin^®^ (Statlab®, Gerwig Ln Columbia, Columbia, MD, USA). Consecutive sections were cut at 2 μm, deparaffinized, hydrated, and submitted to immunohistochemical analysis using the Novolink™ Polymer Detection Systems (Leica Biosystems^®^, Vista, CA, USA) kit, according to the manufacturer’s instructions. Information regarding the primary antibodies and antigen retrieval recovery methods used in this study is summarized in Table 3.

The antibodies were selected to confirm the pluripotent and mesenchymal origin of the eSM-MSCs’ octamer-binding transcription factor 4 (OCT4), homeobox protein (NANOG), proto-oncogene receptor tyrosine kinase or stem cell factor receptor (c-kit), synovial origin (lysozyme), and non-epithelial origin histogenesis (vimentin). Additionally, pan-cytokeratin (AE1 and AE3), synaptophysin, CD31, and glial fibrillary acidic protein (GFAP) were used to confirm there were no vascular, epithelial, neuronal, and neuroendocrine origins of cells, respectively. For each antibody, appropriate negative and positive controls were included, and all primary antibodies were incubated overnight.

The final step consisted of microscopic cell observation, evaluation, and photograph using the microscope Eclipse E600 (Nikon^®^, Tokyo, Japan) and the software Imaging Software NIS-Elements F Ver4.30.01 (Laboratory Imaging^®^, prague, mmun republic). A semi-quantitative score was used for mmunoexpression evaluation, consisting of the percentage of labeled cells (<5%, 5–80%, and >80%) and labeling intensity (0, negative; +, weak; ++, moderate; and +++, strong). Immunoreactivity was considered positive when distinct nuclear and cytoplasmic staining was recognized in at least 5% of the cells. 

### 2.7. eSM-MSC Solution Preparation

The eSM-MSC solution for local clinical application in the 16 equine patients, was a combination of allogenic eSM-MSCs suspended in autologous serum. Prior to preparation of the final therapeutic combination, autologous serum was isolated from whole blood. Then, 10 mL samples of whole blood were collected into dry blood collection tubes, and after clotting, they were centrifuged at 2300 rpm for 10 min and their supernatant (serum) collected and transferred to a 15 mL falcon. Then, the serum was inactivated through a water bath at 56 °C for 20 min followed by cooling on ice. Finally, the serum was centrifuged and filtered using a 0.22 µm syringe filter and stored at −20 °C until further use. Cryopreserved P3 eSM-MSC batches were thawed in a 37 °C water bath, and the content was transferred to a 10 mL tube with autologous serum and slowly diluted, followed by the addition of sterile DPBS until reaching 10 mL. Then, the mixture was centrifuged at 1600 rpm for 10 min. The supernatant was discarded, and the cell pellet was re-suspended in a mixture of autologous serum at a ratio of 0.8:1. Cell counting and viability were determined by using the trypan blue exclusion dye assay (Invitrogen ^TM^, Waltham, MA, USA) using an automatic counter (Countess II FL Automated Cell Counter, Thermo Fisher Scientific^®^, Waltham, MA, USA). Then, the cell number was adjusted to 5 × 10⁶ cells/mL, and then 2 mL of the solution of eSM-MSCs suspended in autologous serum was transferred to a perforable capped vial and preserved on ice until the time of administration.

### 2.8. Treatment Protocol

Twenty structures, tendons and ligaments, were treated with a mixture of allogenic eSM-MSCs and autologous serum. The same treatment protocol was used in every case. All equine patients were submitted to identification, anamnesis, physical examination (cardiac and respiratory frequency, body temperature, mucous membrane examination, inspection of the whole body, and palpation), orthopedic examination (evaluation of the limbs, gait inspection and movements (walk, trot and gallop), and flexion test of the main joints for 60 s followed by trot). Lameness was evaluated at a walk and a trot on hard surface and scored on a scale from 0 to 5, according to the AAEP parameters. Complementary diagnostic exams included regional nerve blocks (to identify the pain area), radiographs, and ultrasound image as reported in other studies [21,24,25,27,28,29,30,31,32].

Following the assumptions of the exclusion criteria, the horses did not receive any treatment before or after the administration of the therapy protocol. In the case of adverse events occurring, such as inflammatory/anaphylactic reactions or infections, the horses should be immediately evaluated and treated with anti-inflammatories or antibiotics, in accordance with their clinical status. The equine patients were monitored in the 48 h after treatment and any occurrences were registered. Following the treatment, the equine patients were assessed periodically to control the equine patient’s healing evolution and to provide valid comparative data among equine patients within the same study group. Table 4 presents the lesion type casuistic.

#### 2.8.1. Intralesional eSM-MSC Injection

Selected horses were sedated with detomidine (0.02 mg/kg), trichotomized, a regional nerve block was performed with lidocaine 2% (20 mg/mL, 2 mL/point), and the surgical skin was disinfected with chlorohexidine and alcohol. The therapeutic combination was aspired to a 2 mL syringe and homogenized, ultrasound was used to identify the lesion site, and an ultrasound guided injection was performed at the lesion over three different points. Finally, a bandage was applied to the limb. All equine patients were injected with phenylbutazone (2.2 mg/kg, IV, SID) at the end of the treatment. The established protocol included a second eSM-MSC administration 15 days after the first treatment using the same protocol.

#### 2.8.2. Clinical Evaluation—Serial Evaluations

Tissue regeneration was estimated through a lameness evaluation, pain to pressure test, limb inflammation, sensitivity, and ultrasound image (reduction of hypoechoic area and fiber alignment). Lesion ultrasonographic evaluations were performed using a 7.5 MHz linear transductor probe (Sonoscape A5^®^, Shenzhen New Industries Biomedical Engineering Co Ltd., Shenzhen, China). For each assessment, a complete examination of the structure was conducted by means of longitudinal and transverse scans. The obtained images were evaluated at each examination for two parameters: lesion echogenicity and lesion longitudinal fiber alignment (FA). The contralateral healthy limb was used as comparison. The evaluation was performed on the treatment day (Day 1) as well as on Days 15, 30, and 45 post-treatments, as presented in Figure 3. According to the classification proposed by Guest et al., this is a short term period study [33]. 

The rehabilitation program consisted of an exercise-controlled program with stall confinement and increasing the amount of time for exercise. Early mobilization included weight-bearing activities, strengthening, and flexibility, and stall rest alone was used as infrequently as possible, as presented on Table 5 [34,35,36,37,38]. Regular ultrasound evaluations were also performed. 

## 3. Results

### 3.1. eSM-MSC Isolation

eSM-MSCs were successfully isolated from equine synovial membrane samples and the average total number of cells isolated from the samples was 1.2 × 10^5^ and 5.6 × 10^5^ at Days 6 and 11, respectively, and expanded from the donor. Cells were observed radiating from the explants and those identified in culture showed clear plastic adherence and mostly fibroblast-like morphology, an essential feature to characterize cells as MSCs (Figure 4a,b).

### 3.2. eSM-MSC Characterization

#### 3.2.1. Tri-lineage Differentiation

Tri-lineage differentiation was confirmed (Figure 5).

##### Adipogenic Differentiation—Oil Red O Staining

Adipogenic differentiation was confirmed by the presence of large red stained lipid vacuoles in the cytoplasm due to exposure of oil red O staining.

##### Chondrogenic Differentiation—Alcian Blue Staining

Chondrogenic differentiation was confirmed by the presence of proteoglycans’ marked deposition in the extracellular matrix which stained blue, confirming the presence of chondrogenic aggregates.

##### Osteogenic Differentiation—Alizarin Red Staining

Osteogenic differentiation was demonstrated by the presence of extracellular calcium deposits stained red by alizarin red solution, which dyes chelate complexes with calcium.

#### 3.2.2. Karyotype Analysis

The cytogenetic analysis revealed the presence of 36% normal cells in P4 and 32% normal cells in P7. Tetraploidy was present in 4% of P4 cells and 8% of P7 cells. Aneuploidy represented 60% of the cells in both passages, hypoploidy being the most representative (56%), as shown at Table 6 and Figure 6.

#### 3.2.3. Secretome Analysis

The analysis of CM revealed the production and secretion of several factors with immunomodulatory functions, capable of intervening beneficially in tissue regeneration. The results of the eSM-MSC CM analysis are shown in Figure 7. Seven biomarkers were identified: keratinocyte chemoattractant/growth regulated oncogene (KC/GRO), monocyte chemoattractant protein-1 (MCP-1), interleukin-6 (IL-6), fibroblast growth factor (FGF-2), granulocyte colony-stimulating factor (G-CSF), granulocyte macrophage colony-stimulating factor (GM-CSF), and interleukin-8 (IL-8). The most expressive were KC/GRO and MCP-1. 

#### 3.2.4. Immunohistochemistry

The eSM-MSCs showed strong expressions of OCT4/NANOG, vimentin, and lysozyme which confirmed marked stem cells, non-epithelial cells, and synovial cells, respectively; weak expression of GFAP; and no expression of CD31, synaptophysin, and pan-cytokeratin, as seen in Figure 8, which confirmed no vascular, neuronal, and epithelial origins of cells. Except for GFAP, in which a smaller number of cells exhibited weaker cytoplasmic immunolabeling in P3 as compared with in passage P0, there was preservation of immunoexpression of all the antibodies between passages P0 and P3. The combination of the positive and negative expressions of these different markers confirmed the expected mesenchymal origin of the cells. Figure 8 presents the immunolabeling of the eSM-MSCs.

### 3.3. Treatment Results

No horse had any adverse event that required study cessation, unplanned procedures, or additional treatments. All intra-tendinous injections and follow-up procedures had no adverse reactions (inflammation, infection, deterioration of the lesion, increased lameness), as shown by Godwin et al. (2012) [39]. No horse had abnormalities identified on the weeks following the injection.

Tendon/ligament regeneration occurred in a time frame of less than 30 days in 80% of the cases and between 30–90 days in 20% of the cases. In this study, eight horses had a lesion on the right front limb, six horses had a lesion on the left front limb, and two horses had a lesion on the right hind limb. There were 14 acute cases and two chronic cases. Chronic cases were diagnosed 6 months before our approach.

After Day 90, meaning they had completed the proposed rehabilitation physical program, the horses started cantering and started to return to their usual work plan. By Day 120 post the first treatment, 87.5% of the horses were back to full work, with the exception of the 12.5% who needed another 30 days to return to full work. 

All horses returned to the same level of sport activity they had before injury. Table 2 and Table 7 summarize the recovery progress, with the respective ultrasound images in Figure 9 and Figure 10. At Day 30, the group that fully recovered demonstrated both a fulfilled ultrasound cross-sectional area and good fiber alignment. There was also no evidence of pain and lameness. Below, transversal and longitudinal ultrasound images of four cases on Day 1 and on Day 30 are presented. After the eSM-MSC treatment, all horses were submitted to a rehabilitation program, as explained in Table 4.

Radiograph exams were performed to rule out the presence of other associated pathologies and regional nerve blocks were performed to better localize the injured region originating the pain.

Ultrasound images at Day 1 and at Day 30 clearly illustrate the evolution of tendon regeneration. Changes in echogenicity, fiber alignment, and cross-sectional area are evident, as seen in Figure 10.

## 4. Discussion

Recently, eSM-MSCs have become an interesting subject for those who study cellular and cell-based therapies due to their promising ability to promote tissue regeneration with high capacity of regeneration of articular structures, tendons, and ligaments. Regarding the collection, isolation, expansion, freezing, and thawing protocols used in this clinical trial, it was possible to use these cells in equine tendon regenerative treatments. The full characterization of eSM-MSCs presents a significant challenge since eSM-MSCs are not as well studied as MSCs from other species, namely human MSCs. However, in this study, their stemness and origins were confirmed through different processes: trilineage differentiation, karyotype, secretome, and immunohistochemistry. All the SM-MSC cultures presented monolayer culture, plastic adherence capacity, and fibroblast-like shape [40,41,42,43], accomplishing some of the minimal criteria defined by ISCT. Successful osteogenic, chondrogenic, and adipogenic differentiation was also demonstrated. De Bari et al. [9] were the first group of researchers to isolate MSCs from synovial tissues.

The karyotype presented some genomic variations when the number of passages was increased. That was consistent with some studies regarding genomic variations along cell passages [44,45,46,47,48]. DNA replication is a critical event for timely genome duplication. Errors in replication lead to genomic instability across evolution [49]. Prieto Gonzalez et al. considered that genomic instability, incurred during the process of stem cell isolation, culture expansion, and reprogramming, might be the most critical point of a stem cell-based therapeutic approach as a viable option from the clinical perspective [50]. Peterson et al. highlighted that there was very little evidence linking genomic abnormalities, for example, in human pluripotent stem cells (hPSCs) with tumorigeneses [44]. The frequency and effects of variations have increased with the development of even more sensitive methods for detecting genomic variation [45].

As reported by Simona Neri, the interpretation of genetic instability and senescence of cultured MSCs is controversial, but the increasing incidence of genetic alterations at advanced culture times clearly indicates that few culture passages correspond to a reduced chance to harbor dangerous alterations. Therefore, prudent behavior is desirable with a reduction in culture times as much as possible to avoid safety concerns [51]. More studies must be performed in this area. 

During the last decade, it has been shown that the therapeutic effectiveness of MSCs is due mainly to the release of paracrine factors, namely CM, composed of soluble (cytokines, chemokines, and growth factors) and nonsoluble factors (extracellular vesicles) that are primarily secreted in the extracellular space by stem cells [52]. CM’s paracrine signaling can be considered to be the primary mechanism by which MSCs contribute to the healing process, and therefore, their study has become an interesting subject [53,54].

In our study, eSM-MSCs revealed a CM with a high level of KC/GRO, MCP-1, Il-6, FGF-2, G-CSF, GM-CSF, and IL-8. This highlights the intense activity of fibroblasts, producing KC/GRO that is chemotaxic for neutrophils during inflammation. MCP-1 is essential for reperfusion and the successful completion of musculoskeletal tissue after an ischemic injury [55]. Macrophages are tissue resident cells involved in tissue regeneration along with their inflammatory and infection responses [56]. IL-6 is a proinflammatory and angiogenic interleukin capable of increasing the expression of growth factors; reactivating, for example, intrinsic growth programs of neurons; promoting axonal regrowth; and creating a link between inflammation and tissue regeneration [57,58]. FGF-2 is a recognized GF responsible for proliferation of tenogenic stem cells. FGF-2 signaling has been reported to produce a tendon progenitor population that expressed scleraxis during somite development [59]. FGF-2 plays a crucial role in cell proliferation and collagen production, becoming a useful GF for tissue regeneration by promoting stem cell proliferation [60]. G-CSF is a cytokine that mobilizes bone marrow-derived cells (BM-DCs) to peripheral blood. A study suggested that injection of G-CSF to promote BM-DC release in the target area, i.e., rotator cuff, effectively enhanced rotator cuff healing by promoting tenocyte and cartilage matrix production [61]. Wright et al. presented a study that confirmed skeletal muscle damage, including muscle damage following strenuous exercise, induced an elevation in plasma G-CSF, implicating it as a potential mediator of skeletal muscle repair [62]. Recent human trials have shown the benefits of G-CSF administration as a treatment for neuromuscular diseases, considering that G-CSF affects skeletal muscle, leading to functional improvements [63,64,65,66,67,68]. GM-CSF is an hematopoietic growth factor with proinflammatory functions [69]. Major sources of GM-CSF are T and B cells, monocyte/macrophage endothelial cells, and fibroblasts. Neutrophils, eosinophils, epithelial cells, mesothelial cells, Paneth cells, chondrocytes, and tumor cells can also produce GM-CSF [70]. Paredes et al. evidenced that elevated levels of proinflammatory factors such as those found at these cells CM (GM-CSF, G-CSF, Il-6, IL-8 and IL-17), were implicated in the activation of resident tendon cells for effective healing, stimulating tendon cell proliferation [71,72]. IL-8 is one of the major mediators of inflammatory response and is a potent angiogenic factor. This is similar to IL-6, but IL-8 has a longer half-life [73].

A recent study highlighted that hematopoietic factor promoted tendon healing in aged mouse tendons. Histochemical results demonstrated that vascularization of the injury site was significantly elevated. It was concluded that vascular endothelial growth factor (VEGF) played an important role in decreasing adipocyte accumulation and also improved vascularization of the tendon during aged tendon healing. Active regulation of VEGF may improve the treatment of age-related tendon diseases and tendon injuries [74].

Studies with human BM-MSCs using a human-specific proteome profiler array with different angiogenic factors such as VEGF-A, IL-6, IL-8, platelet-derived growth factor A (PDGF-A), endothelin-1 (ET1), and urokinase plasminogen activator (uPA), which had not been previously reported in the CM of human MSCs, were also identified in an equine array, confirming what we found in this study [75]. This factor has been proposed as a modulator of the different neovascularization stages, through the enhancement of VEGF gene promotor activity [75,76]. Schokry et al. [77] reported that BM-MSC therapies have recovery times of 3–6 months and conservative therapeutic methods allow recovery in 12–18 months without regeneration but with formation of fibrous scar tissue. Retrospectively, no re-injuries of tendons have occurred in horses treated with this new approach, during the study frame time. In the literature [78], Smith et al. referred to a low percentage re-injury rate of 27% for SFD tendonitis treated with bone marrow stem cells. Horses returned to “full function” as defined by Cook et al. and modified by Guest et al. [33,79].

A study using a murine osteoarthritis (OA) model demonstrated that an injection of MSCs CM, similarly to injection of MSCs, resulted in early pain reduction and had a protective effect on the development of cartilage damage in a murine OA model, by using the regenerative capacities of the MSCs-secreted factors [80].

Interestingly, the results accumulated so far have provided evidence that veterinary patients affected by naturally occurring diseases should provide more reliable outcomes of cell therapy than laboratory animals, thus, allowing translating potential therapies to the human field. More recently, a cell-free therapy based on MSCs CM has been proposed. Even though there are very few clinical reports to refer to in veterinary medicine, recent acquisitions suggest that MSC-derived products may have major advantages compared to the related cells, for example, they are considered safer and less immunogenic [52]. As evidenced before, eSM-MSC CM factors are able to promote tendon healing by reducing inflammation and fatty infiltration, stimulating cell proliferation and tenogenic differentiation [81].

In this study we used a cell-based therapy instead of CM itself, but we were aware of its effect and potential on cell-based therapies; its advantages and therapeutic effects were the reason why this study was performed.

To better characterize the cells under study, we performed immunohistochemistry assays. The choice of markers was based on a previous work [8] and included several of the criteria used for humans, as determined by the ISCT. Results of our study demonstrated the presence of the embryonic stem cell markers OCT4 and NANOG. Detection of these markers has been previously described by Beltrami et al., in multipotent adult stem cells (HMASC) from human bone marrow [82], as well as, by Riekstina et al., who also demonstrated the presence of these markers in HMASC derived from bone marrow, adipose tissue, heart, and dermis [83]. Greco et al. also evidenced elevated expression of OCT4 in P3 MSCs and hypothesized OCT4 expression could be an indicator of MSC differentiation potential in clinical diagnostics [84]. In equine characterization of synovial fluid and membrane-derived MSCs, Prado et al. also evidenced the presence of NANOG and OCT4 markers [19]. In contrast, Fulber et al. had no positive results for these two markers in equine mesenchymal stem cells of synovial tissues [43]. Vimentin, a mesenchymal stem cell marker, was also detected, suggesting the mesenchymal origin of cells. The presence of lysozyme confirmed the synovial origin of cells, as stated by Fulber et al. [43].

The immunohistochemistry analysis showed the absence of CD31, sinaptophysine, and pan-cytokeratin expressions, confirming no vascular, neuronal and epithelial origins of cells. GFAP was weakly expressed, being less expressed in P3 than in P0 cells. CD31 was performed to investigate the presence of hematopoietic cells in eSM-MSCs. The expression of VEGF was not found, these results being similar to those from Fulber et al., and to other authors that evidenced the absence of hematopoietic markers [43,85]. The absence of neuronal and dermal markers was also consistent with other studies [19,43].

In our clinical trial, we treated mainly early acute lesions; 87.5% of the cases were acute lesions of tendons or ligaments. Therefore, we created a master cell bank of allogenic eSM-MSCs suitable for treatments in early acute phases versus treatments with autologous cells where time of tissue collection, preparation, and cell culture need to be considered. Furthermore, cell harvesting for autologous treatment is an invasive procedure which is unnecessary with this new eSM-MSC solution. The possibility of having a master cell bank enables faster healing of the organ and a quicker return to sport life. Horses spend less time in recovery time and have a regenerated tissue instead of a fibrotic tissue. These are some advantages of the eSM-MSC solution. Another concern is that in the early stages of the lesion there is an inflammatory phase; however, the paracrine factors released by eSM-MSCs also have anti-inflammatory action, reducing inflammation. 

Chronic cases represented 12.5% of the cases, involving four structures. Three of the horses recovered in 30 days and one of the horses had a delayed recovery time.

The delayed recovery time in 20% of the structures, meaning 12.5% of the horses, was due to, in Case 6, an increased number of involved structures (more than one tendon or ligament) and a foot conformation abnormality, as the horse had a fetlock hyperextension that was impairing correct tendon healing. This was corrected with special shoeing. Inappropriate rehabilitation program (Case 7) was another cause of delayed recovery time. As soon as the corrective shoeing was performed, ligament regeneration started.

We could also conclude that lameness grade was not directly correlated with lesion cross-sectional area. Horses with ultrasonographic cross-sectional grade 1, 2, and 3 lesions presented lameness grade 4/5, which was observed in 9 of 16 patients. Lameness grade 3/5 was presents in 4 of 16 of equine patients with ultrasonographic cross-sectional grade 1 and 2 lesions. Lameness grade 2/5 was present in 3 of 16 equine patients with ultrasonographic cross-sectional grade 1 lesions.

Kamm et al. (2021) concluded that based on the evidence to date, tendons appear to have improved healing when treated with allogeneic MSCs, and the use of these treatments in equine tendon and ligament lesions is warranted [86]. Colbath et al. (2020) claimed that some of the advantages of using allogenic stem cells include the ability to bank cells and to also reduce the treatment time, to collect MSCs from younger donor animals, and the ability to manipulate banked cells prior to administration [87]. Some of the disadvantages focused on the risk of immunological reactions. However, currently, there are several studies in horses accumulating evidence that allogeneic MSCs may be a safe alternative to autologous MSCs [87]. Nevertheless, the donor’s health must always be taken into consideration as well as the donor’s age [88].

## 5. Conclusions

To sum up, this study accomplishes the criteria for reporting veterinary and animal medicine research for MSCs in orthopedic applications [33] and the ISCT perspective on immune assays for MSC’s criteria for advanced phase clinical trials [89], confirmed by plastic adherence, tri-lineage differentiation, synovial membrane origin, spindle-shaped cells, as well as proliferative and immune modulatory capacity proven by immunohistochemistry and CM. 

From a clinical point of view, the idea of having an allogenic eSM-MSC cell bank is very interesting. Therefore, the possibility of having a universal donor who can provide a large amount of eSM-MSCs, to culture and preserve non-immunogenic cells whose availability is immediate, allowing a quick and effective therapeutic answer in acute stages of musculoskeletal lesion is the paramount goal of orthopedic medicine.

From a “one-health” perspective, equines play an important role as a model for human musculoskeletal disorders; the high-level analogy between human and equine structures may have a great translational value for both species for future clinical aspects [28,90]. There are significant resemblances between equine SDFT and human Achilles tendon with respect to the size of anatomical structure and load, function (energy store), pathophysiology of tendon injury, and the healing response under activity or traumatic rupture compared to other species [90]. Moreover, considering the result of tendinopathy in equine species which reflects the conditions encountered in humans, the horse is accepted as an appropriate model in this area by the research community and by other authorities such as the U.S. Food and Drug Administration (FDA) and the European Medicines Agency (EMA).

Based on the clinical, ultrasonographic, and performance outcomes identified in the present study, the use of eSM-MSCs together with autologous serum solution has proven its efficiency for tendon and ligament repair and contributes to reduce the recovery period and subsequent rapid return to athletic activity. The therapy was demonstrated to be safe and had no adverse findings. The clinical results and athletic outcomes of the horses were very positive. Comparing our study with others, using for example BM-MSCs, it seems that our new approach has shorter recovery times and fewer re-injuries [39,77]. These results encourage the use of eSM-MSCs and autologous serum for the treatment of tendonitis and desmitis, since they can regenerate tendon and ligament tissue and regain organ function, enhancing the return to competition in excellent time frames.

## Figures and Tables

**Figure 1 animals-13-01312-f001:**
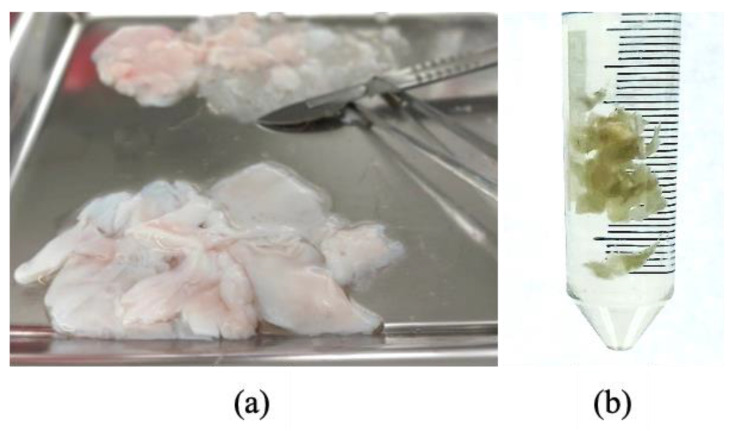
Laboratory arrival and preparation of the fresh tissue to start digestion, isolation, and expansion: (**a**) Tissue collected in the field; (**b**) isolated synovial membrane.

**Figure 2 animals-13-01312-f002:**
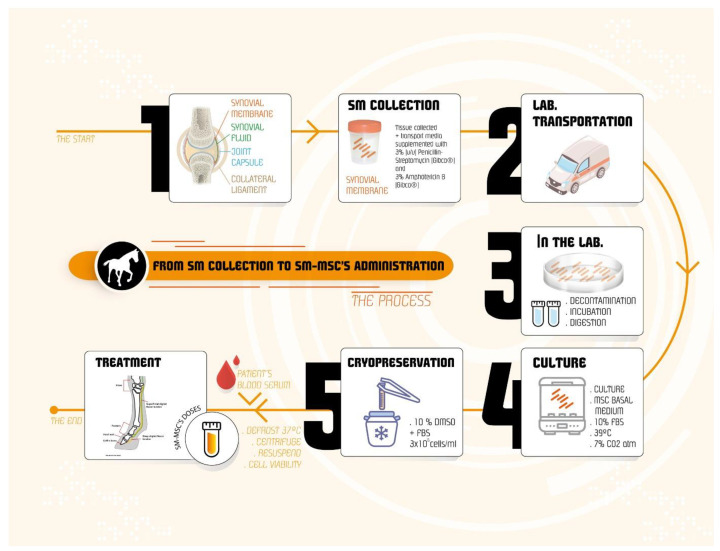
Schematic representation of the event sequence from the collection of synovial membrane to the administration of the therapeutic combination. After the collection, the synovial membrane is transported to the laboratory where it is separated from the whole tissue, decontaminated, incubated, and digested. Then, cells are cultured and expanded and finally cryopreserved in a cell bank. When needed for treatment, cells are prepared with autologous serum, and then applied in the selected equine patient.

**Figure 3 animals-13-01312-f003:**
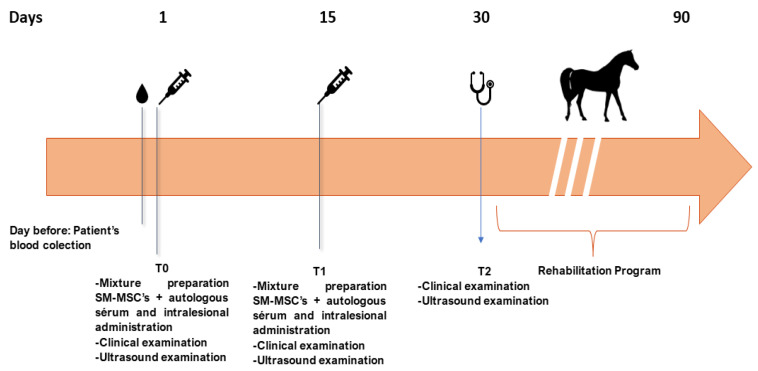
Timeline of the eSM-MSC treatment protocol and rehabilitation program. The day before the first treatment (T0), blood from the equine patient was collected to prepare autologous serum. At T0, the mixture of autologous serum and eSM-MSCs was injected intralesionally after a clinical and ultrasound examination. After 15 days, the same procedure was repeated. At day 30 (T2), a clinical and ultrasound examination was performed and if a favorable outcome was considered, the horse progressed to a physical rehabilitation program. During the physical rehabilitation program, the equine patient was also re-evaluated at days 60 and 90.

**Figure 4 animals-13-01312-f004:**
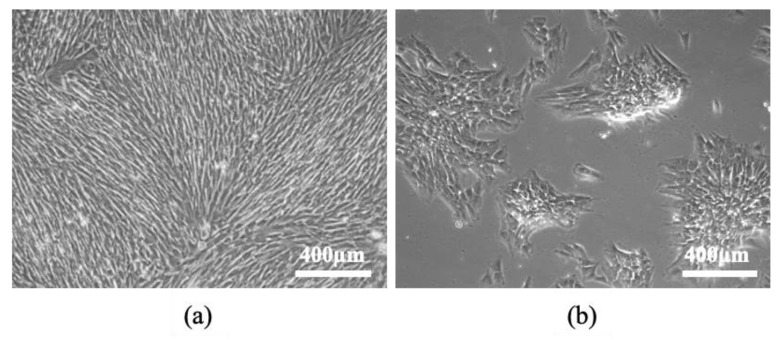
eSM-MSCs in culture, isolated through enzymatic digestion: (**a**) Passage 0 (P0); (**b**) Passage 1 (P1). Plastic adherence, monolayer, and fibroblast-like shape of eSM-MSCs may be observed. Magnification, 100×.

**Figure 5 animals-13-01312-f005:**
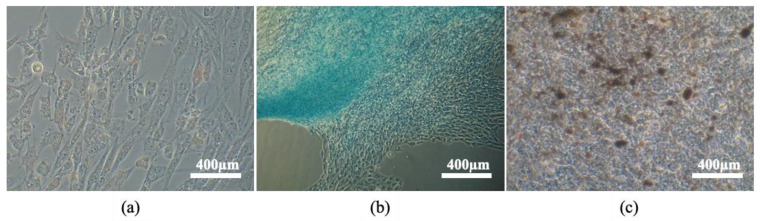
Tri-lineage differentiation: (**a**) eSM-MSC’s adipogenic differentiation, cytoplasmatic lipid vacuoles stained in red (Oil red staining); (**b**) eSM-MSC’s chondrogenic differentiation, proteoglycans in extracellular matrix stained in blue (Alcian blue staining); (**c**) eSM-MSC’s osteogenic differentiation, extracellular calcium deposits stained in red (alizarin red staining). Magnification 100×.

**Figure 6 animals-13-01312-f006:**
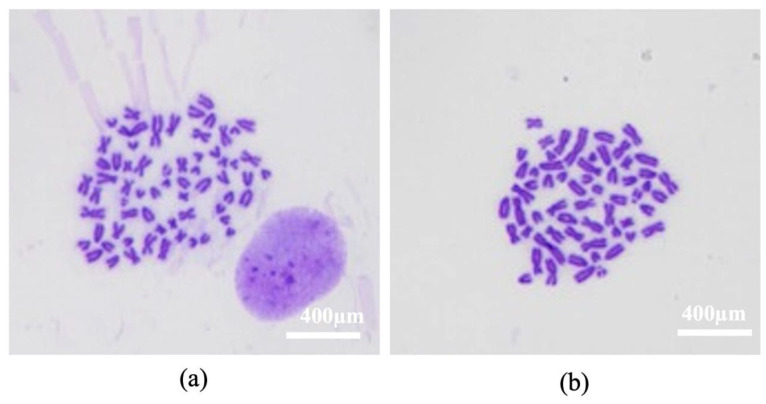
Karyotype. Images of eSM-MSC cytogenetic analysis Passage 7 (P7): (**a**) Normal karyotype, 64 chromosomes, XY; (**b**) hypoploid cell, 59 chromosomes, 3 acro and 2 submeta. Magnification 100×.

**Figure 7 animals-13-01312-f007:**
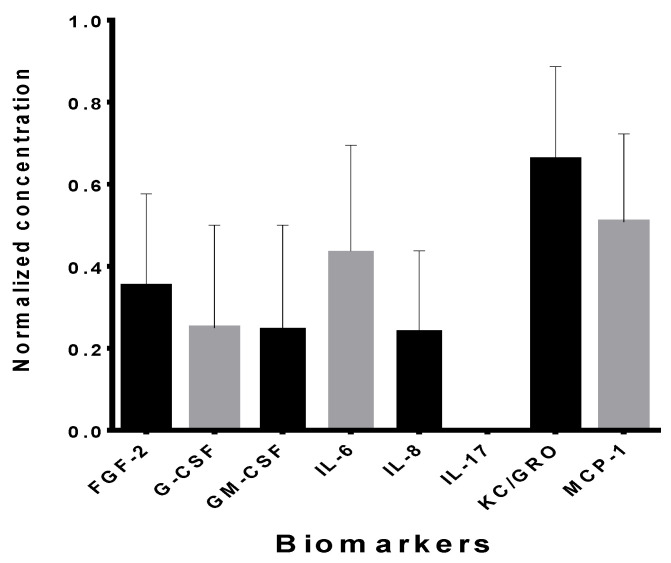
Biomarkers, normalized concentration of each biomarker in the conditioned medium of eSM-MSCs P4 (mean ± SEM).

**Figure 8 animals-13-01312-f008:**
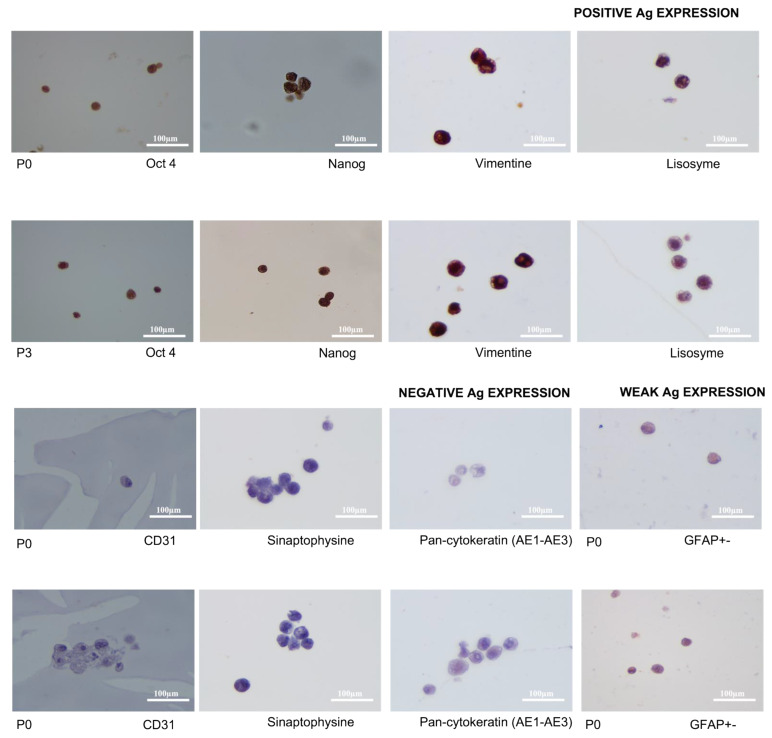
Immunolabeling of eSM-MSCs P0 and P3. Magnification 600×. Images present positive Ag expression of OCT4 and NANOG confirming stem cells, positive expression of vimentin confirming non-epithelial cells, and positive expression of lysozyme confirming synovial cells. Positive expression was revealed by cytoplasmatic staining of the cells. CD31, synaptophysin and pan-cytokeratin had negative expressions, did not stain, and confirmed no vascular, neuronal, or epithelial origins of cells. GFAP represents a neuronal origin of cells and had a weak expression in P0, which reduced in P3. Magnification 600×.

**Figure 9 animals-13-01312-f009:**
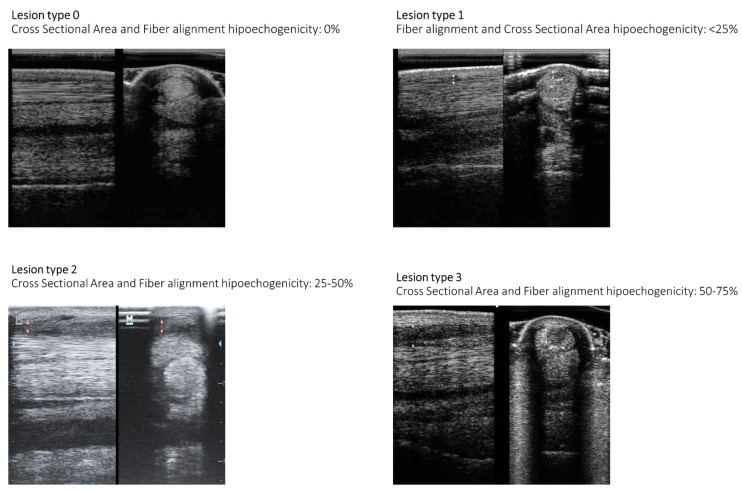
Illustration of ultrasonographic lesion characterization summarized in Table 7. Longitudinal fiber alignment and cross-sectional area echogenicity loss is presented [27].

**Figure 10 animals-13-01312-f010:**
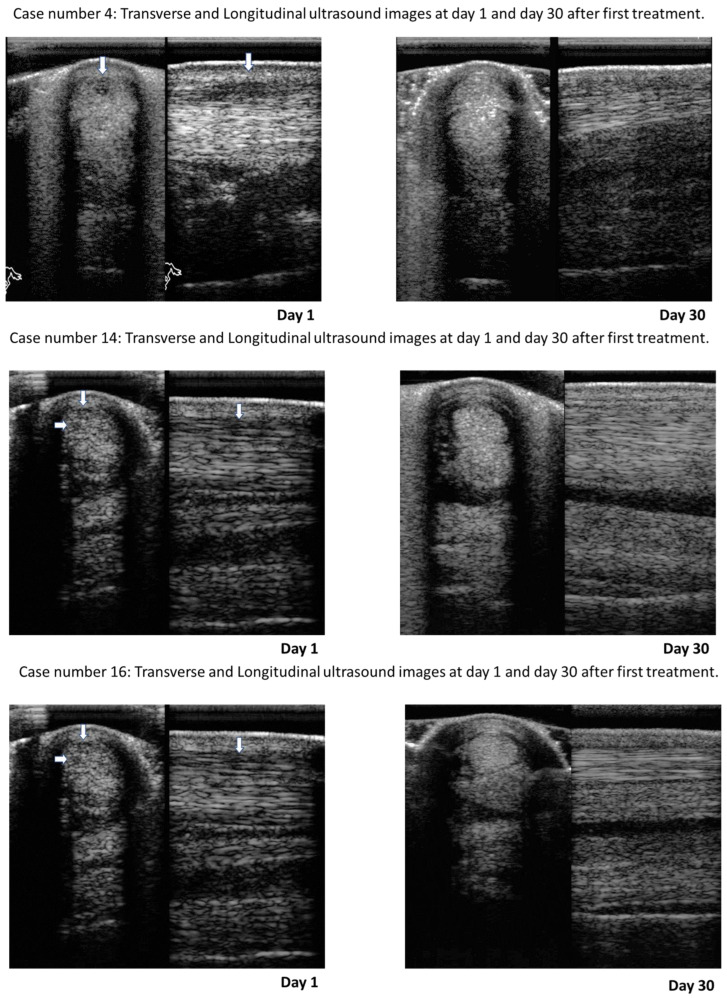
Ultrasound images of Case numbers 4, 14, and 16 that represent the clinical cases concerning superficial digital flexor tendon (SDFT). Transversal and longitudinal images at Day 1 and at Day 30 after first treatment with eSM-MSCs. These ultrasounds are representative of the cases and very illustrative of good fiber alignment and cross-sectional area reduction, evidencing tissue regeneration.

**Table 1 animals-13-01312-t001:** Score systems used by the veterinary surgeon to assess lameness, and responses to flexion and pain to pressure tests [25,26].

Parameter	Score	Clinical Implication
AAEP Grading	0	No Lameness
	1	Lameness not consistent
	2	Lameness consistent under certain circumstances
	3	Lameness consistently observable on a straight line.
	4	Obvious lameness at walk: marked nodding or shortened stride
	5	Minimal weight bearing lameness in motion or at rest
Flexion Test	0	No flexion response
	1	Mild flexion response
	2	Moderate flexion response
	3	Severe flexion response
Pain to pressure	0	No pain to pressure
	1	Mild pain to pressure
	2	Moderate pain to pressure
	3	Severe pain to pressure

**Table 2 animals-13-01312-t002:** Equine patient and lesion characterization. The left column characterizes the equine patients: Sex, male (M) or female (F); age measured in years old (yo); sports modality (SM), show jumping (SJ), dressage (Dre), and Leisure (Lsr); lameness score (AEEP score) pretreatment. The right column characterizes the lesions: Structure affected, superficial digital flexor tendon (SDFT), deep digital flexor tendon (DDFT), and suspensory ligament (SL) left branch (LB); affected limb, right frontlimb (RF), right hindlimb (RH), left frontlimb (LF) and left hindlimb (LH).

Equine Patient					Lesion: Tendonitis/Desmitis			
ID	Sex	Age (yo)	SM	Lameness Pretreatment	Structure	Type	Limb	Evolution
1	M	22	SJ	3/5	SDFT DDFT	Acute	LF	Favorable in 30 days
2	F	5	SJ	2/5	SDFT	Acute	RH	Favorable in 30 days
3	F	14	SJ	2/5	SDFT	Chronic	RF	Favorable in 30 days
4	M	8	SJ	4/5	SDFT	Acute	RF	Favorable in 30 days
5	M	7	SJ	4/5	LB SL	Acute	LF	Favorable in 30 days
6	M	13	Lsr	4/5	SDFT DDFT SL	Chronic	RH	Tendons: favorable evolution in 30 days; SL in 90 days.
7	M	15	SJ	4/5	SFDT	Acute	RF	Favorable in 90 days
8	M	11	SJ	4/5	SDFT	Acute	RF	Favorable in 30 days
9	F	10	SJ	4/5	SDFT SL	Acute	RF	Favorable in 30 days
10	M	9	SJ	4/5	SDFT	Acute	LF	Favorable in 30 days
11	F	10	SJ	3/5	SDFT	Acute	RF	Favorable in 30 days
12	M	12	Dre	2/5	SDFT	Acute	LF	Favorable in 30 days
13	M	14	SJ	4/5	SL	Acute	LF	Favorable in 30 days
14	M	7	SJ	4/5	SDFT	Acute	RF	Favorable in 30 days
15	M	12	SJ	3/5	SFDT	Acute	LF	Favorable in 30 days
16	F	6	SJ	3/5	SFDT	Acute	RF	Favorable in 30 days

**Table 3 animals-13-01312-t003:** List of antibodies investigated, dilutions, and antigen retrieval methods applied in the immunohistochemical analysis. Oct 4—Octamer-binding transcription factor 4; NANOG—Homeobox protein NANOG; GFAP—Anti-Glial Fibrillary Acidic Protein; CD31—Platelet endothelial cell adhesion molecule.

Marker	Type/Clone	Supplier	Dilution/Incubation Period	Antigen Unmasking	Positive Control	Cells of Interest	Reference
OCT4	Polyclonal	Abcam	1/100 ON	RS/WB	Canine mast cell tumor	Stem cells	Ab18976
NANOG	Clone Mab	ABGENT	1/10 ON	RS/WB	Canine testicular carcinoma	Stem cells	AM1486b
c-Kit (CD117)	Polyclonal	Dako Denmark	1/450 ON	RS/WB	Canine mast cell tumor	Stem cells	A4502
Lysozyme	Polyclonal	Dako Denmark	1/400 ON	RS/WB	Canine synovial membrane	Synovial cells	A0099
Vimentin	Clone V9	Dako Denmark	1/500 ON	RS/WB	Canine mammary gland	Non-epithelial cells	M0725
Pan-cytokeratin	Cocktail AE1/AE3	Thermo Scientific	1/300 ON	RS/WB	Canine mammary gland	Epithelial cells	M3-343-P1
GFAP	Polyclonal	Merck Millipore	1/2000 ON	RS/WB	Mouse brain tissue	Neuronal cells	AB5804
Sinaptophysine	Clone SP11	Thermo Scientific	1/150 ON	RS/WB	Mouse brain tissue	Neuronal cells	RM-9111-S
CD31	Clone JC70A	Dako Denmark	1/50 ON	Pepsine	Canine spleen	Platelet endothelial cells	M0823

**Table 4 animals-13-01312-t004:** Lesion type casuistic.

Lesion Type	No. Clinical Cases	Total Number (2019)
Tendonitis	16	20
Desmitis	4	

**Table 5 animals-13-01312-t005:** Physical rehabilitation program. After eSM-MSC treatment, all horses were submitted to a rehabilitation program consisting of two days of box rest followed by 13 days of 10 min of hand walking. The bandage applied on treatment day was removed 24 h after treatment. At Day 15, the second treatment was performed followed by another 15 days of rehabilitation, until Day 30. Between Days 30 and 45, the work consisted of 20 min hand walking; between Days 45 and 60, the work was 30 min of hand walking; between Days 60 and 75, the work was 30 min of hand walking plus 5 min trotting; and finally, between Days 75 and 90, each horse was submitted to 30 min of hand walking plus 10 min of trotting. After this, the horses could return to full work.

Week	Exercise
0–2	Stall confinement, 2 days Hand walking, 10 min Day 15, new treatment
3–4	Stall confinement, 2 days Hand walking, 10 min VET-CHECK on Day 30
5	Hand walking, 15 min
6	Hand walking, 20 min VET-CHECK on Day 45
7	Hand walking, 25 min
8	Hand walking, 30 min VET-CHECK on Day 60
9–10	Hand walking, 30 min + 5 min trotting
11–12	Hand walking, 30 min + 10 min trotting VET-CHECK on Day 90

**Table 6 animals-13-01312-t006:** Cytogenetic analysis in Passages 4 and 7 (P4 and P7). Percentage of normal cells, tetraploid cells, and aneuploid cells.

P4	Cytogenetic Analysis	P7
36%	Normal cells 64, XY	32%
4%	Tetraploid cells 128 XXYY	8%
60%	Aneuploid cells:	60%
	Hipoploidy 54–63	
56%		56%
	Hiperploidy 71	
4%		4%

**Table 7 animals-13-01312-t007:** Ultrasonographic lesion characterization at Days 1, 15, and 30. Equine patient identification, structure affected, lesion ultrasonographic location, cross-sectional area, and longitudinal fiber pattern are characterized. Assessment outcome is also evaluated. Affected structure superficial digital flexor tendon (SDFT), deep digital flexor tendon (DDFT), suspensory ligament (SL), and left branch (LB). Lesion ultrasonographic location (zones 1A-1B, 2A-2B, 3A-3B) [27], cross-sectional area % (0, 0%; 1, <25%; 2, >25–50%; 3, >50–75%; 4, >75%), longitudinal fiber pattern (0, 0%; 1, <25%; 2, >25–50%; 3, >50–75%; 4, >75%) [27], and assessment outcome (full function, acceptable function, and unacceptable function) [33].

Patient ID	Day	Structure	Location	Cross-Sectional Area	Longitudinal Fiber Pattern (%)	Assessment Outcome
1	1	SDFT	1A-1B	1	1	
		DDFT	1A-1B	1	1	
	15	SDFT	1A-1B	1	1	
		DDFT	1A-1B	1	1	
	30	SDFT	1A-1B	0	0	Full function
		DDFT	1A-1B	0	0	
2	1	SDFT	1A-1B	1	1	
	15	SDFT	1A-1B	1	1	
	30	SDFT	1A-1B	0	0	Full function
3	1	SDFT	1A-1B	1	1	
	15	SDFT	1A-1B	1	1	
	30	SDFT	1A-1B	0	0	Full function
4	1	SDFT	1A-1B	2	2	
	15	SDFT	1A-1B	2	2	
	30	SDFT	1A-1B	0	0	Full function
5	1	LB SL	3A-3B	2	2	
	15	LB SL	3A-3B	2	2	
	30	LB SL	3A-3B	0	0	Full function
6	1	SDFT	2A-2B	2	2	
		DDFT	2A-2B	2	2	
		SL	2A-2B	2	2	
	15	SDFT	2A-2B	1	1	
		DDFT	2A-2B	1	1	
		SL	2A-2B	2	2	
	30	SDFT	2A-2B	1	1	Unacceptable function. Only at day 90.
		DDFT	2A-2B	1	1	
		LS	2A-2B	1	1	
7	1	SDFT	2A-2B	3	3	
	15	SDFT	2A-2B	3	3	
	30	SDFT	2A-2B	2	2	Unacceptable function.
8	1	SDFT	2A-2B	2	3	
	15	SDFT	2A-2B	2	2	
	30	SDFT	2A-2B	0	0	Full function
9	1	SDFT	2A-2B	2	1	
	15	SDFT	2A-2B	1	1	
	30	SDFT	2A-2B	0	0	Full function
10	1	SDFT	1A-1B	3	3	
	15	SDFT	1A-1B	2	2	
	30	SDFT	1A-1B	0	0	Full Function
11	1	SDFT	1A-1B	2	2	
	15	SDFT	1A-1B	2	2	
	30	SDFT	1A-1B	0	0	Full function
12	1	SDFT	2A-2B	1	1	
	15	SDFT	2A-2B	1	1	
	30	SDFT	2A-2B	0	0	Full function
13	1	SL	1A-1B	2	2	
	15	SL	1A-1B	1	1	
	30	SL	1A-1B	0	0	Full function
14	1	SDFT	2A-2B	1	1	
	15	SDFT	2A-2B	1	1	
	30	SDFT	2A-2B	0	0	Full function
15	1	SDFT	2A-2B	2	2	
	15	SDFT	2A-2B	2	2	
	30	SDFT	2A-2B	0	0	Full function
16	1	SDFT	2A-2B	2	2	
	15	SDFT	2A-2B	2	2	
	30	SDFT	2A-2B	0	0	Full function

## Data Availability

The data that support the findings of this study are available from the corresponding author on request.

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
