# Peer review of "Allogenic Synovia-Derived Mesenchymal Stem Cells for Treatment of Equine Tendinopathies and Desmopathies—Proof of Concept"

_animals, 2023, doi:10.3390/ani13081312_

Round 1

Reviewer 1 Report

Allogenic synovia derived mesenchymal stem cells for treatment of equine tendinopathies and desmopathies - The Proof- of-concept

Reis, I., Lopes, B., Sousa, P., Sousa, A., Branquinho, M., Caseiro, A., Pedrosa, S., Rêma, A., Oliveira, C., Porto, B., Atayde, L., Amorim, I., Alvites, R., Santos, J., Maurício, A.

Originality/Novelty

The question is well defined. The authors contribute new information.

Significance

The results are interpreted appropriately and are a contribute to the knowledge about allogenic synovia derived mesenchymal stem cells use for the treatment of equine tendinopathies and desmopathies.

All conclusions are justified and supported by the results.

Quality of presentation

The article is written in an appropriate way. Data and analysis are presented appropriately.

Scientific soundness

The study is correctly designed and technically sound and the data seem robust enough for the conclusions drawn.

Generally, the methods, tools, and software, are described with sufficient details to allow another reseacher to reproduce the results.

Nevertheless, the authors need to state in Materials and Methods how they planned to deal with adverse events.

Furthermore, the authors need to improve the quality of the print of Figure 2, and Figure 9 if necessary could be removed..

Interest to the readers

The conclusions are interesting for the readership of the journal.

Overall Merit

The work provides an advance towards the current knowledge and is relevant for the clinical field.

English level

The English language is appropriate and understandable.

REVIEW REPORT

Brief summary

The authors aimed at developing a solution of equine synovial membrane stem cells and autologous serum, to be injected at the lesion site (20 structures – tendons and ligaments –, of 14 superficial digital flexor tendons, 2 deep digital flexor tendons and 4 suspensory ligaments) to promote tissue regeneration. The process of tissue collection, preparation, isolation of synovial membrane mesenchymal stem cells (eSM-MSCs), expansion, culture, cryopreservation and posterior preparation with autologous serum is described. These cells are fibroblast-like and grow in clusters. They retain an osteogenic, chondrogenic and adipogenic differentiation potential. The use of autologous serum as a vehicle aims at contributing to an immunogenic response after the administration of this combination and to its healing properties, due to the presence of growth factors and immunoglobulins. After treatment with allogenic eSM-MSCs and autologous serum, all horses underwent a physical rehabilitation program and were monitored with physical and ultrasonographic exams. The authors report the results to be very promising, as 14/16 healed in 30 days , encouraging the use of equine synovial stem cells and autologous serum in the treatment of tendonitis and desmitis.

Broad comments

This is a prospective longitudinal study Feb. 2016-Jan.2019. Sixteen ambulatory clinic horses with acute and chronic signs of lameness (AAEP scale + regional nerve block + flexion test + pain to pressure). Show jumping (14), dressage (1) and leisure (1).

The authors have addressed a significant topic where there are still some gaps in current knowledge and the present work represents a step forward in tendo/ligament injury healing.

The methods used are generaly well described. The paper is generally well written, but some specific comments pointed out underneath should be addressed.

Specific comments

Line 166 - 5 to 22 years old

Line 187 - established

Line 204 – please specify the age of “foal”

Figure 2 - Some parts are difficult to read (e.g. The text underneath “SM Collection”). Number 3 should be “IN THE LAB”.

Line 221 – please replace “had” by “has”

Line 247 – “in the selected equine patient.”

Line 252 – “cells in P4 were used after thawing .

Line 284 - for 20 minutes

Line 287 – 3% acetic acid

Line 306 – assay

Line 316-318 - After centrifugation (1700 rpm), 8 ml of ice-cold fixative comprising methanol and glacial acetic at a proportion of 3:1, was added and mixed. Afterwards, the cells were centrifuged again.

Line 345 - to new centrifugation cycles

Line 425 - followed by trot

Line 426 - scored on a scale

Line 430 - administration of the protocolled therapy

Line 432 - Following the treatment, patients were assessed periodically to control patient’s healing evolution

Line 466 - Rehabilitation program consisted of an exercise-controlled program with stall confinement and increasing time of exercise.

Line 474 – intralesionally

Line 538 - 36% of normal

Line 540 -    , hypoploidy being the most representative (56%),

Line 609 – Regarding “No horse had any adverse event that required study cessation, unplanned proce- 609 dures or additional treatments.”  - The authors need to state in Materials and Methods how they planned to deal with adverse events.

Line 621 - who needed aniother 30 days to return to full work.

Line 624 – This sentence needes to be refrased: “In the group that fully recovered at day 30, ultrasound cross-sectional area was fulfilled as well as a good fiber alignment”

Line 625 – Below, a transversal and longitudinal ultrasound image of 4 cases on day 1 and day 30 is presented.

Line 629 - to better localize the injured region originating the pain.

Figure 9 – If necessary, could be removed as it is a previously published scoring system (reference 27)

Line 677 - The full characterization

Line 678 - as well studied as

Line 683 -differentiation was also demonstrated

Line 684 - were the first group

Line 685 – The karyotype

Line 692 - highlighted that there is

Line 738 - highlights that hematopoietic factors promote

Line 739 - demonstrated that vascularization of

Line 752 – but with  formation

Line 779 - from human bone marrow

Line 793 - these results being similar to those from … and to other

Line 800 - Furthermore, cell harvesting for autologous treatment is an invasive procedure which becomes unnecessary with this new product.

Line 804 - in the early stages of the lesion there is an inflammatory phase, and the paracrine factors

Line 811 - and a foot conformation abnormality, as the horse had a fetlock hyperextension that was imparing the correct

Line 819 - and to reduce

Line 820 - Some of the disadvantages focused on the risk of immunological reactions.

Line 845 - Moreover, considering the result of tendinopathy in equine species which reflects the conditions encountered in human, the horse is accepted as an appropriate model in this area by the research community and by other…

Line 854 - , it seems that our

RECOMMENDATION - Accept after minor revision.

Author Response

Review Report – Reviewer 1

Dear reviewer 1:

We would like to thank the reviewer´s comments on the manuscript “Allogenic synovia derived mesenchymal stem cells for treatment of equine tendinopathies and desmopathies - The Proof-of- concept ". The manuscript has been modified and improved according to the suggestions of the reviewers. All changes introduced appear in the final document highlighted in yellow.

1) In the Materials and Methods:

- The authors need to state in Materials and Methods how they planned to deal with adverse events.

As suggested by the reviewer, a paragraph was added with the plan in case of adverse events (Line: 425-428).

“In case of adverse events occurrence, such as inflammatory/anaphylactic reactions or infections, animals should be immediately evaluated and treated with anti-inflammatories or antibiotics, in accordance with their clinical status.”

2) Figures:

- Improve the quality of figure 2 and 9.

The quality of figure 2 and 9 was improved.

3) Specific comments

The authors agree with the Reviewer comments and corrections from the specific comments, so all the respective lines were altered, following the Reviewer’s suggestions.

Reviewer 2 Report

The manuscript is well written and presents a very rich detailing of data and results. I believe that rare and minor flaws do not detract from its publication. Some information I really missed:

Improved description of the donor and the respective joint. How was the joint/synovial health confirmed?

Classify the severity of tendonitis and its association with clinical signs (lameness). In the manuscript there is a separation between acute and chronic tendonitis, but there is no relationship between percentage of tendon or ligament injury and lameness score pretreatment. How did the chronic lesions appear? There is no description.

Does phenylbutazone affect healing/treatment results?

The response to treatment of chronic or acute injuries can be quite different, depending on how the injuries were classified.

Author Response

Review Report – Reviewer 2

Dear reviewer 2:

We would like to thank the reviewer´s comments on the manuscript “Allogenic synovia derived mesenchymal stem cells for treatment of equine tendinopathies and desmopathies - The Proof-of- concept ". The manuscript has been modified and improved according to the suggestions of the reviewers. All changes introduced appear in the final document highlighted in yellow.

How did the chronic lesions appear? There is no description.

As suggested by the reviewer, a paragraph was added explaining how the chronic lesion appear (Line: 186-191).

In this study, horses with acute or chronic lameness, with diagnosed tendonitis and/or desmitis and with no signs of systemic disease were accepted in the inclusion criteria. Injured horses were treated in acute stages of disease, “except for two patients – number 3 and number 6. Patient number 3 had an injury the year before of this treatment and did laser therapy. Never had a complete recovery. After that, he had a reinjury and at this time, this treatment was suggested. Patient number 6 was referred by another clinician who tried, unsuccessfully, to treat this patient. Patient was sent to the field for one year and then reevaluated. At this time, and as its tutor wanted to improve its life quality, this treatment was proposed by its clinician. Lameness grade of each patient is specified at table 6.”

Improved description of the donor and the respective joint.

How was the joint/synovial health confirmed?

As suggested by the reviewer, a paragraph was added to improve the donor description, as well as the respective joint (Line:  210-214).

eSM-MSCs’ donor was a young and healthy foal, 7 months old, who died accidentally when running in the arena. The tutor authorized synovial membrane collection from hocks, knees, and fetlocks. Synovial membrane was evaluated and its appearance - transparent, bright and smooth, as well as the presence of viscous and transparent synovial fluid confirmed its soundness.”

In the manuscript there is a separation between acute and chronic tendonitis, but there is no relationship between percentage of tendon or ligament injury and lameness score pretreatment.

As suggested by the reviewer, the information was added in table 6 (Line:  644-645).

Table 6.  Equine patient and Lesion characterization.

Left column characterizes Equine patient:  Sex - male (M) or female (F); age measured in years old (yo); sports modality- SM: Show jumping (SJ), Dressage (Dre) and Leisure (Lsr); Lameness score (AEEP score) pretreatment. Right column characterizes Lesion: Structure affected - Superficial Digital Flexor Tendon (SDFT), Deep Digital Flexor Tendon (DDFT) and Suspensory Ligament (SL) – Left branch (LB); affected Limb - Right Frontlimb (RF), Right Hindlimb (RH), Left Frontlimb (LF) and Left Hindlimb (LH).

Equine Patient

Lesion: Tendonitis / Desmitis

ID

Sex

Age

(yo)

SM

Lameness Pre treatment

Structure

Type

Limb

Evolution

1

M

22

SJ

3/5

SDFT

DDFT

Acute

LF

Favorable in 30 days

2

F

5

SJ

2/5

SDFT

Acute

RH

Favorable in 30 days

3

F

14

SJ

2/5

SDFT

Chronic

RF

Favorable in 30 days

4

M

8

SJ

4/5

SDFT

Acute

RF

Favorable in 30 days

5

M

7

SJ

4/5

LB SL

Acute

LF

Favorable in 30 days

6

M

13

Lsr

4/5

SDFT

DDFT

SL

Chronic

RH

Tendons: favorable evolution in 30 days; SL in 90 days.

7

M

15

SJ

4/5

SFDT

Acute

RF

Favorable in 90 days

8

M

11

SJ

4/5

SDFT

Acute

RF

Favorable in 30 days

9

F

10

SJ

4/5

SDFT

SL

Acute

RF

Favorable in 30 days

10

M

9

SJ

4/5

SDFT

Acute

LF

Favorable in 30 days

11

F

10

SJ

3/5

SDFT

Acute

RF

Favorable in 30 days

12

M

12

Dre

2/5

SDFT

Acute

LF

Favorable in 30 days

13

M

14

SJ

4/5

SL

Acute

LF

Favorable in 30 days

14

M

7

SJ

4/5

SDFT

Acute

RF

Favorable in 30 days

15

M

12

SJ

3/5

SFDT

Acute

LF

Favorable in 30 days

16

F

6

SJ

3/5

SFDT

Acute

RF

Favorable in 30 days

Classify the severity of tendonitis and its association with clinical signs (lameness). 

As suggested by the reviewer, a paragraph was added to classify the severity of tendonitis and its association with clinical signs (Line:  824-829).

“We could also conclude that lameness grade was not directly correlated with lesion cross-sectional area. Horses with ultrasonographic cross-sectional lesions grade 1,2 and 3 presented lameness grade 4/5, which was observed in 56.25% of the cases.  Lameness 3/5 represented 25% of the cases and appeared in patients with ultrasonographic cross-sectional lesions grade 1,2.  Lameness 2/5 represented 18.75% of the cases and patients with ultrasonographic cross-sectional lesions grade 1.”
